# Uncertainty-Aware Deep Learning Classification of Adamantinomatous Craniopharyngioma from Preoperative MRI

**DOI:** 10.3390/diagnostics13061132

**Published:** 2023-03-16

**Authors:** Eric W. Prince, Debashis Ghosh, Carsten Görg, Todd C. Hankinson

**Affiliations:** 1Department of Neurosurgery, University of Colorado School of Medicine, Aurora, CO 80045, USA; 2Department of Biostatistics and Informatics, Colorado School of Public Health, Aurora, CO 80045, USA; 3Morgan Adams Foundation Pediatric Brain Tumor Research Program, University of Colorado School of Medicine, Aurora, CO 80045, USA

**Keywords:** deep learning, brain tumor diagnosis, uncertainty quantification, craniopharyngioma

## Abstract

Diagnosis of adamantinomatous craniopharyngioma (ACP) is predominantly determined through invasive pathological examination of a neurosurgical biopsy specimen. Clinical experts can distinguish ACP from Magnetic Resonance Imaging (MRI) with an accuracy of 86%, and 9% of ACP cases are diagnosed this way. Classification using deep learning (DL) provides a solution to support a non-invasive diagnosis of ACP through neuroimaging, but it is still limited in implementation, a major reason being the lack of predictive uncertainty representation. We trained and tested a DL classifier on preoperative MRI from 86 suprasellar tumor patients across multiple institutions. We then applied a Bayesian DL approach to calibrate our previously published ACP classifier, extending beyond point-estimate predictions to predictive distributions. Our original classifier outperforms random forest and XGBoost models in classifying ACP. The calibrated classifier underperformed our previously published results, indicating that the original model was overfit. Mean values of the predictive distributions were not informative regarding model uncertainty. However, the variance of predictive distributions was indicative of predictive uncertainty. We developed an algorithm to incorporate predicted values and the associated uncertainty to create a classification abstention mechanism. Our model accuracy improved from 80.8% to 95.5%, with a 34.2% abstention rate. We demonstrated that calibration of DL models can be used to estimate predictive uncertainty, which may enable clinical translation of artificial intelligence to support non-invasive diagnosis of brain tumors in the future.

## 1. Introduction

Deep learning (DL) imaging models are a widely explored Artificial intelligence (AI) method to predict brain tumor diagnosis from magnetic resonance imaging (MRI) [1,2,3,4,5,6,7]. Expected benefits include reduced diagnostic error and cost, better treatment planning, and better patient education [8,9,10,11,12,13,14]. With their ability to recognize patterns and identify key features in medical images, DL models can provide an efficient and accurate means of diagnosing brain tumors. Jaju et al. offer a comprehensive resource on the utility of pediatric brain tumor images and a survey of interpretation concepts, thus providing valuable insights for designing AI systems to support clinicians [15]. Similarly, Xie et al. conducted a comprehensive review of 83 studies from 2015–2022, focusing on the use of DL models for the diagnosis of brain tumors from MRI, reporting an accuracy range of 60–100% [16]. Neither review included suprasellar tumors such as adamantinomatous craniopharyngioma (ACP).

Few AI technologies have been adopted in the clinical setting due to the “black-box” problem of DL: its scale and complexity make it difficult to understand, leading to a lack of trust in the technology among clinical decision-makers [17,18]. In addition to the lack of clarity in the exact algorithmic process of a DL classification, there is also no information provided regarding the predictive uncertainty of the classification. Predictive accuracy is a measure of correctness, while predictive uncertainty represents the reliability of the accuracy measurement. It is critical to know the predictive uncertainty of a DL model in a high-risk scenario, such as neuro-oncology, because errors can have devastating effects [19]. Studies have shown that, prior to adopting AI-based decision support, clinicians require information regarding the reliability of model performance and what kind of mistakes it can make [20,21,22]. Under high uncertainty, the model should also be able to state “*I don’t know*” and abstain from providing a prediction [23].

Probabilistic deep learning (DL) methods have been used to create more generalizable and effective models that capture parameter variability and report predictive certainty. For instance, Shamsi et al. proposed a DL framework to detect COVID-19 using medical images of chest X-ray and CT scans [24]. Four DL models (VGG16, ResNet50, DenseNet121 and InceptionResNetV2) were used to extract deep features, which were then processed by machine learning and statistical models to identify cases of the virus. The linear support vector machine and neural network models were found to have the highest accuracy, sensitivity, specificity, and AUC. The predictive uncertainty estimates of CT images were also found to be higher than those of X-ray images; however, the method utilized by the authors only measured the uncertainty of the training data and did not evaluate uncertainty on a held-out test set. Another example is Hercules, a DL-based model for medical image classification that features an uncertainty-aware module and achieves high accuracy in retinal, lung, and chest imaging datasets [25]. This method outperforms existing state-of-the-art medical image classification methods. It utilized a specialized architecture followed by a Markov chain sampling procedure to estimate the predictive uncertainty. In contrast, our approach is unique in the way it specifically constructs a calibration curve rather than estimates the predictive uncertainty without constraint.

Clinical experts can accurately diagnose pediatric suprasellar brain tumors, such as ACP, from preoperative MRI with 86% accuracy, but neurosurgical biopsy is still required for 91% of diagnoses [26,27]. Radiographic diagnosis offers a non-invasive solution that avoids risks and costs associated with neurosurgical intervention. However, radiographic interpretation can be limited by low spatial resolution of radiology images and atypical visual features which make diseases look similar [28]. This limitation then requires neurosurgical biopsy and pathological examination for a definitive diagnosis [28,29]. Radiographic interpretations can affect the accuracy of a pathological diagnosis and treatment planning, so it is important to maximize certainty [28]. DL can detect brain tumors from radiology images and is sensitive to minor image alterations that are not perceptible to humans [7,30]. We hypothesize that AI can be utilized to improve the reliability of radiographic diagnosis by supporting the clinician when they are uncertain.

We present a novel method that uses Bayesian DL to predict ACP diagnosis from preoperative MRI. Specifically, our method calibrates a deep learning image classifier for preoperative MRI of ACP tumors in a one versus all context for pediatric suprasellar tumors. We also investigate if the mean and variance of the predictive distribution is informative to flag when a prediction is uncertain, and we present an algorithm which can abstain from making a prediction when uncertain.

## 2. Materials and Methods

### 2.1. Image Acquisition

For this research, we utilized the same dataset that was used in our prior study [7]. This multi-institutional dataset is comprised of sagittal T1-weighted MRI comprised of 39 ACP patients and 47 patients, which have other suprasellar tumors (i.e., NOTACP). These included pilocytic astrocytoma (n=12), germinoma (n=7), pilomixoid astrocytoma (n=6), optic glioma (n=4), pituitary adenoma (n=3), arachnoid cyst (n=3), prolactinoma (n=3), mature teratoma (n=2), low grade glioma (n=2), renal cell carcinoma (n=2), Rathke’s cyst (n=1), lipoma (n=1), and Langerhans cell histiocytosis (n=1). Three representative two-dimensional images were extracted from each patient by a board-certified neurosurgeon. For training, 23 ACP and 30 NOTACP patient datasets were used, with three representative images per patient and imaging modality (six images per patient, 318 images total). All training images were augmented to synthetically expand the dataset to 1000 images (500 ACP, 500 NOTACP). The test dataset consisted of 16 ACP and 17 NOTACP patients, with one image selected per patient and imaging modality (n=33).

### 2.2. Deep Learning Classification of Preoperative MRI Images and Benchmarking

The DL computations were performed with the TensorFlow framework, including the TensorFlow Hub, TensorFlow Probability, and TensorFlow AddOns libraries [31]. The images were first preprocessed by the ResNet v2 preprocessing function (Figure 1a). A ResNet v2 152 model, which had been pre-trained on the ImageNet dataset, was used to produce a 2048-dimensional feature embedding for each input image [32]. These embeddings were used to train a DL classifier or a benchmarking model (i.e., random forest or XGBoost).

In our previous work, we used a genetic algorithm to optimize the DL model architecture and meta-parameters for training [7]. The final model included batch normalizing the feature embeddings (with a batch size of two) and then passing them through a dropout layer, which randomly masks 50% of the feature values before they go to the hidden fully connected layer with a softmax activation function. The batch normalization and final dense layer are the only elements of the architecture with optimizable parameters, and only the dense layer is calibrated for uncertainty (Figure 1a).

Our new approach optimizes the parameters using a LazyAdam optimizer, a decoupled weight-decay, a categorical cross-entropy objective function, cyclic learning rate, and smoothed via stochastic weight averaging [33,34,35,36]. The LazyAdam optimizer is designed to handle sparse updates, which regularizes model weights to focus on rare and distinguishable features, leading to better training loss and generalization error. By adjusting the weight decay factor independently from the learning rate for LazyAdam, the generalization performance is improved. This combined approach is based on the idea that imaging features that commonly occur are assigned a lower learning rate, while uncommon features are focused on with higher learning rate. As a result, the optimizer enables the discovery and distinguishing of highly predictive yet uncommon features, allowing the classifier to discern between highly similar instances (e.g., suprasellar tumors). The performance was assessed using categorical accuracy and area under the precision-recall curve (AUPR).

### 2.3. Bayesian Subspace Inference

Uncertainty can be divided into aleatoric (inexplicable due to randomness) and epistemic (reducible with more/better data or better understanding of the knowledge) uncertainty [37,38]. Point estimation for parameters in DL models is one source of reducible epistemic uncertainty; a non-Bayesian DL model is trained to find a single set of optimal parameter values given observations of x and y, but Bayesian methods account for multiple possible solutions [39,40,41]. Calculating multiple solutions can be intractable for DL models with billions of parameters, but approximations of the distributions can be calculated [42]. For example, the parameter space can be mapped through principal component analysis (PCA) to create a simple linear subspace that represents the complicated solution space, which enables us to construct a geometric curve that connects potential solutions and is suitable for Bayesian DL predictions [42].

We applied the subspace inference method in [42] to construct a calibrated induced posterior of our deep learning model, as illustrated in Figure 1b. This process consists of three steps. First, we constructed a low-dimensional subspace of the parameter space using the first principal components of the optimizer trajectory on the loss surface. Next, we used a geometric curve to connect two high performing models in the constructed subspace. Lastly, we applied elliptical slice sampling to sample from the geometric curve, yielding predictive distributions of the suprasellar tumor diagnosis. To implement this approach, we utilized TensorFlow (available at https://github.com/lericnet/bsi-acp-mri (accessed on 30 January 2023) to support our existing model. The approach is described in further detail in the following subsections.

#### 2.3.1. Construct PCA Subspace Curve End-Point Parameters through SWA Training

We tuned two separate models with different random seeds (6567 and 7656) for the calibration curve in Section 2.3.2. To identify optimal anchor points, we utilized stochastic weight averaging (SWA) to smooth the models’ tuning process [33]. Following Algorithm 2 in [42], every 10 training steps (i.e., epochs), the model parameters are averaged, and a deviation matrix (the error between the current parameters and the 10 epoch average) is calculated. This bias matrix is then decomposed via PCA and is stored as a model parameter. SWA is a technique used in deep learning to improve the generalization of a model. It involves taking the average of a neural network’s weights obtained from multiple training cycles, rather than just the weights at the end of the last training cycle. This averaging technique reduces overfitting and provides a more robust model with better generalization performance. Furthermore, our use of a cyclic learning rate enables SWA to explore regions that are rich in potential solutions [33], resulting in two DL models that are optimized to encourage generalization. We refer to these models as SWA-ResNet.

#### 2.3.2. Connecting Loss Modes by Bezier Curves in PCA Subspace

Given the two resulting models, the next step was to connect these two optima through a learned PCA subspace representation [40]. Briefly, each SWA-ResNet model serves as a fixed end-points (${\hat{w}}_{1}$ and ${\hat{w}}_{2}$) and the free center parameter θ is optimized such that the curve is fit across a high accuracy space [40]. The final 3-point quadratic Bezier curve ($\phi_{\theta }$) is generated using the equation:$$\phi_{\theta }\left(t\right)={\left(1-t\right)}^{2}{\hat{w}}_{1}+2t\left(1-t\right)\theta +t^{2}{\hat{w}}_{2},\;\;0\leq t\leq 1$$
where t is the position along the length of the curve. A Bezier curve is a parametric curve, which is constructed via interpolation between two specified points. The curves are symmetric and can be reliably transformed through PCA space to reliably map to the more complex DL parameter space. Importantly, the Bezier curve exists on what is known as the convex hull of the parameter space; this the minimum curve, which contains the solutions between our defined end points and all interior points. This means that we will return, and efficient representation of our parameter subspace will be used for variational inference.

#### 2.3.3. Variational Inference by Elliptical Slice Sampling

The final step of the process is to calculate the posterior distribution of the model. We sample from the posterior parameters in our induced PCA subspace and then transform these parameters back to the original parameter space. The selection of an inference procedure is a matter of experimental design, and we have a broad range of approximate inference methods to choose from [42]. We utilize elliptical slice sampling (ESS)—a Markov chain Monte Carlo method of performing inference in models with multivariate Gaussian priors—implemented through the TensorFlow Probability library [43]. These sampled values are passed as the weights for the dense layer in our model, resulting in 5000 (ACP, NOTACP) paired predictions.

### 2.4. Computational Infrastructure

All computational programs were performed on 64-core Intel Gold 6226R CPUs with 512 GB RAM, and 128 GB of NVIDIA GPU memory running CentOS 7.4.1708. Visualizations were generated using custom R scripts supported by the tidyverse ecosystem [44].

## 3. Results

### 3.1. SWA-Tuned Model Outperforms Benchmarks

The first step of the approach is to identify two DL classifiers with parameters identified from wide valleys of the PCA subspace landscape. Figure 2 shows ResNet feature embeddings of test MRI images, which were best classified by the SWA-tuned classifier (AUPR = 0.871), improving performance beyond our previously published model (AUPR = 0.808). The most common machine learning classifiers deployed are random forest and XGBoost. Both our previously published model and SWA-tuned model outperforms random forest and XGBoost in terms of accuracy and AUPR. AUPR is utilized because we care more about identifying the positive class (ACP) than the negative class (NOTACP) because the former is clinically meaningful diagnosis.

### 3.2. PCA Subspace Model Results in Stable but Reduced Test Performance 

Given the two SWA tuned models, we constructed a three-point curve with fixed endpoints (Figure 3a). Our curve model was constrained to only update the center-point value, θ. We observed that our weight parameter endpoints remained fixed during training epochs and that the center-point was modified, resulting in a flatter distribution of weight parameters. Mapping the curve points through PCA subspace resulted in approximately normally distributed curve weights, $\phi_{\theta }\left(t\right)$, with more than four times the variance of ${\hat{w}}_{1}$, θ, or ${\hat{w}}_{2}$ (Figure 3b). This curve is what we utilize to generate our predictive distributions; we can utilize variational inference methods (i.e., ESS) to sample weights from the curve which are passed to the Dense classification layer, resulting in a distribution of predictions for each input image. Therefore, the increase in variance in curve weights versus ${\hat{w}}_{1}$, θ, or ${\hat{w}}_{2}$ indicates that our model is more uncertain. During our curve-fitting process, this was corroborated by a stable test performance while training, but AUPR was lower for this model than SWA-tuned and our prior model (Figure 3c). Note, the test performance at this stage is utilizing the mean value from the curve weights. In the next section, we sample from the curve weights to generate predictive distributions for each patient.

### 3.3. Mean Prediction Values Are Ambiguous, but Variance of Prediction Distributions Is Informative

For each test patient MRI (n=33), we generated conditional probability distributions for the ACP and NOTACP classes (Figure 4a; red and blue, respectively) using elliptical slice sampling (n=5000). Aside from one patient (the last row of the ACP patients), there are no major observable differences in the mean values between prediction classes. For example, given the first patient, the mean value for NOTACP is only marginally greater than the mean value for ACP. A similar trend can be seen throughout the test dataset. Next, we evaluated whether the difference between each of the class mean probabilities could be used to identify uncertain predictions (Figure 4b). Filtering predictions based on a p-value threshold resulted in no changes in predictive performance in terms of sensitivity, specificity, false negative rate, false positive rate, and accuracy. As we increased the threshold for abstaining from prediction, we only observed a loss in cases predicted with no benefit in performance. In total, this indicates that the difference in mean values for ACP/NOTACP predictive distributions is not immediately indicative of the predictive uncertainty of our classifier. 

Although Figure 4a shows that the means do not appear significantly different, there are some patterns with respect to variance across the two classes; the boxplots in the ACP cohort (Figure 4a, top half) typically have wider ranges (Figure 4c,d) than those in the NOTACP cohort (Figure 4a, bottom half). This indicates that the model is more generally uncertain about predicting ACP patients than NOTACP patients. We sought to leverage this observation and calculated the statistical significance of a given patient belonging to a ground truth group based on their predictive variance using a Kolmogorov-Smirnov (KS) test. Plotting the p-values for each class (Figure 4e), we observed that the ACP and NOTACP classes (circles and triangles, respectively) could be separated by the ratio of p-values (Figure 4e, diagonal line). Therefore, the variance of the predictive distributions is informative in our classification task.

### 3.4. Classification Algorithm with Abstention Mechanism Improves Test Performance

We developed an algorithm that combines ArgMax predictions, as well as predictions based on the variance of the ESS prediction, and the steps of the algorithm are shown in Figure 5. ArgMax (top-left) selects the class with the highest predicted value. For example, the ArgMax prediction of (0.75 ACP, 0.25 NOTACP) is ACP. This approach on the calibrated model resulted in an accuracy of 68.6%, sensitivity of 0.882, specificity of 0.5, false negative rate (FNR) of 0.118, and a false positive rate (FPR) of 0.5. The second prediction method (top-right) is based on the NOTACP/ACP ratio of KS p-values; ratios greater than 1 are predicted as NOTACP, while those that are less than 1 are predicted to be ACP. This approach resulted in an accuracy of 85.7%, sensitivity of 0.882, specificity of 0.833, FNR of 0.118, and FPR of 0.167.

The sensitivity and FNR was unchanged between the two prediction methods. We then compared the predictions from each method and observed a high accuracy if both methods predict the same value. Specifically, we observed an accuracy of 95.5%, sensitivity of 0.933, specificity of 0.833, FNR of 0.067, and FPR of 0.125. This was in the context of abstaining from prediction in 34.2% (n=12) of cases. Out of the abstained cases, 10/12 (83.3%) were from the NOTACP group; 2/12 (16.6%) were from the ACP group.

## 4. Discussion and Conclusions

We demonstrated that Bayesian DL can be used to calibrate a deep learning image classifier to predict ACP diagnosis from preoperative MRI. Statistical analysis of the predictive distributions revealed that the variance of the distribution could be leveraged to trigger the model to abstain from prediction, which we implemented in a two-step diagnostic algorithm.

We observed that model performance on the test images varied across each stage of our study. At first, the SWA-tuned classifier (AUPR = 0.871) performed better in classifying ResNet feature embeddings of test MRI images than our previously published model (AUPR = 0.808) [7] and outperformed the other models. However, our final calibrated model only achieved an accuracy of 68.6%. These results demonstrate that the prediction is now reflecting the model’s uncertainty, depending on the data it has seen; subspace inference is especially valuable for indicating increasing or decreasing uncertainty when making new predictions relative to the data it has already seen [27]. Given the limited amount of data, there is a high probability of overfitting, which implies that the model may not be applicable to new images. We utilized multiple techniques, such as transfer learning, dropout, data augmentation, and stochastic weighted averaging, to limit overfitting. Nevertheless, inference through the PCA subspace showed that the model was still overfit, which indicates the necessity of calibrating DL models in the context of small training data and high-risk application scenarios, such as healthcare. By calibrating our model, we likely have increased the model’s ability to be applicable to other data sets, but more research is necessary to investigate generalizability.

Calibration of our DL model using this approach improved specificity and revealed that the uncertainty is predominantly associated with the negative (NOTACP) class. Two reasons support this outcome: first, we specifically optimized for performance based on AUPR, which emphasizes the positive class, ACP; second, the NOTACP class is ambiguous compared to the ACP class, since ACP is comprised of one diagnosis, while NOTACP is comprised of eleven diagnoses (see the Methods Section). It is important to note that a radiographic differential diagnosis is a ranked list of potential diagnoses given the radiographic images and known clinical information about the patient [45]. Within this concept, there is the prediction of the diagnosis and the prediction of the diagnosis rankings. We are specifically addressing the prediction of the diagnosis here. In future work, we plan to expand the approach to also address the prediction of the diagnosis rankings and generate a more realistic radiographic differential diagnosis by including pituitary adenoma, craniopharyngioma, meningioma, astrocytoma, germinoma, as well as non-cancerous pathologies, such as aneurysm.

Our approach has several limitations that could be addressed in future work. The sample size of our training and test data is relatively small and could be increased to improve generalizability. Additionally, our study utilized two-dimensional images, rather than volumetric MRI, which is more commonly used in a clinical setting to interpret neuroimaging. Using volumetric MRI would likely improve accuracy of the model. We could also explore the application of other inferential methods. Our current method, elliptical slice sampling, is not an exact solution of the posterior integral [43], and, therefore, the uncertainty measurement is an approximation. As an alternative solution, we could evaluate the probably approximately correct (PAC)-Bayes approach and more computationally expensive methods, such as Hamiltonian Monte Carlo [46,47,48], for exact solutions. Simpler solutions, such as dropout, could also be utilized as an approximation of the Bayesian model average. These different approximations and exact solutions of the posterior integral could be compared and evaluated in terms of computational complexity, stability, and cost.

Uncertainty is only one of the remaining challenges that need to be addressed for successfully translating AI to the clinic. Another critical component to successfully translating such a model to the clinic will be reliant on the usability, or accessibility, of the model’s prediction and uncertainty with respect to the clinical user. To date, very few studies have focused on clinical usability of AI system [16]. Considering how the prediction and uncertainty are presented to the clinician and interacted with are vitally important to the design of a clinical AI system. Future work will investigate how different visual design interfaces impact the usability and utility of our model system.

In conclusion, the presented approach holds advantages for the design of clinical AI tools to support diagnosis of pediatric brain tumors, such as ACP. The variance of the predictive distributions can inform when the predicted diagnosis is uncertain. Furthermore, when the prediction is considered certain, the diagnostic accuracy is beyond that reported for human experts. Current research is underway to expand this concept to calibrate a multi-class classifier to run as an orthogonal prediction to the one-versus-all approach in this study. Applying a multi-class classifier will create a prediction that is represented by a list of brain tumor diagnoses ranked by decreasing likelihood combined with a prediction for each brain tumor diagnosis versus all others. Both the ranked list and one-versus-all classifiers can be calibrated using the presented approach. As a result, our method provides a basis to use DL for predicting pediatric brain tumor diagnosis from preoperative MRI with desired uncertainty quantification properties using Bayesian methods on a calibrated PCA parameter subspace.

## Figures and Tables

**Figure 1 diagnostics-13-01132-f001:**
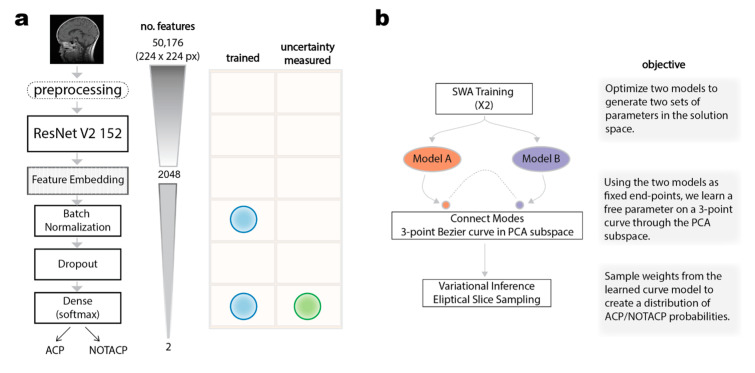
Overview of deep learning model architecture (**a**) and calibration approach (**b**).

**Figure 2 diagnostics-13-01132-f002:**
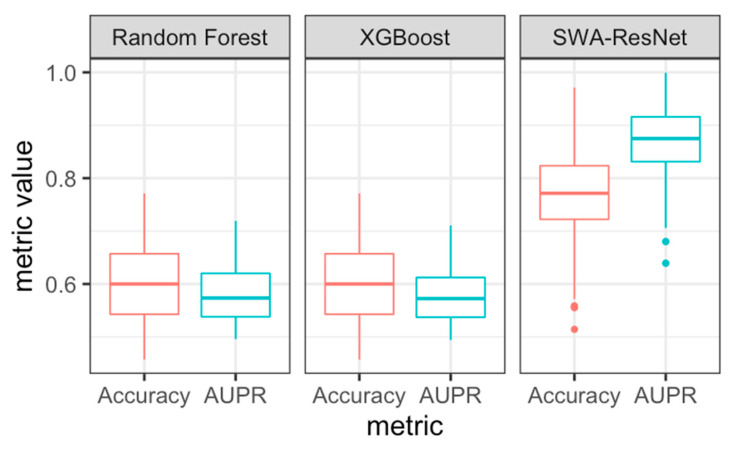
Comparison of accuracy (as a decimal, red boxplot) and area under the precision-recall curve (AUPR, blue boxplot) of classifiers on ResNet feature embeddings of ACP MRI.

**Figure 3 diagnostics-13-01132-f003:**
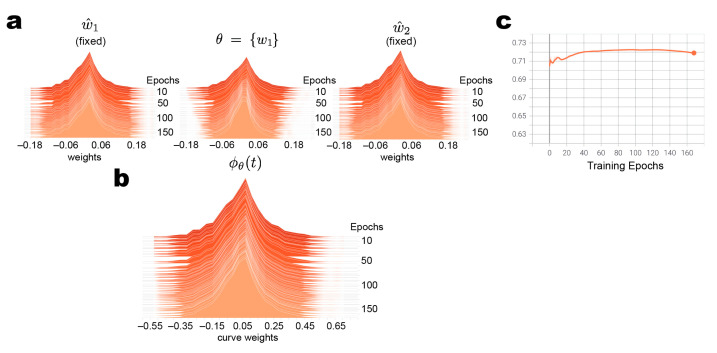
PCA subspace model parameter distributions and test performance. (**a**) Parameter distributions during curve-fitting (during training epochs) showing the fixed endpoints (outside distributions) and the optimized parameter. (**b**) Distribution of parameters within calibrated curve. (**c**) Test performance (AUPR) during curve-fitting shows stable performance during the optimization of the free parameter θ.

**Figure 4 diagnostics-13-01132-f004:**
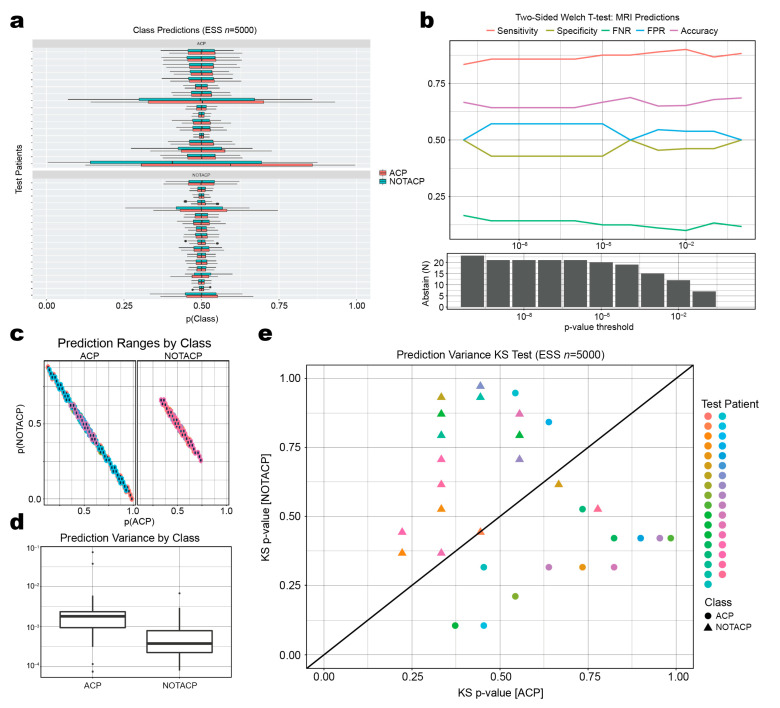
Prediction distributions with mean and variance relationships. (**a**) Predictive distributions (generated using elliptical slice sampling; ESS) stratified by patient ground truth (top APC, bottom NOTACP) and predicted class (red = ACP, blue = NOTACP). (**b**) Model performance using mean values for abstention across *p*-value threshold. (**c**) Predictive distributions for each diagnostic class, stratified by ground-truth. Colors indicate different patients prodived in the legend in (**e**). (**d**) Boxplot of predictive distribution variance stratified by ground-truth. (**e**) Scatterplot of Kolmogorov-Smirnov *p*-values for each predicted class. Each point represents one test patient, ACP/NOTACP ground-truth depicted as circles and triangles, respectively. Diagonal line represents classification boundary used in abstention algorithm.

**Figure 5 diagnostics-13-01132-f005:**
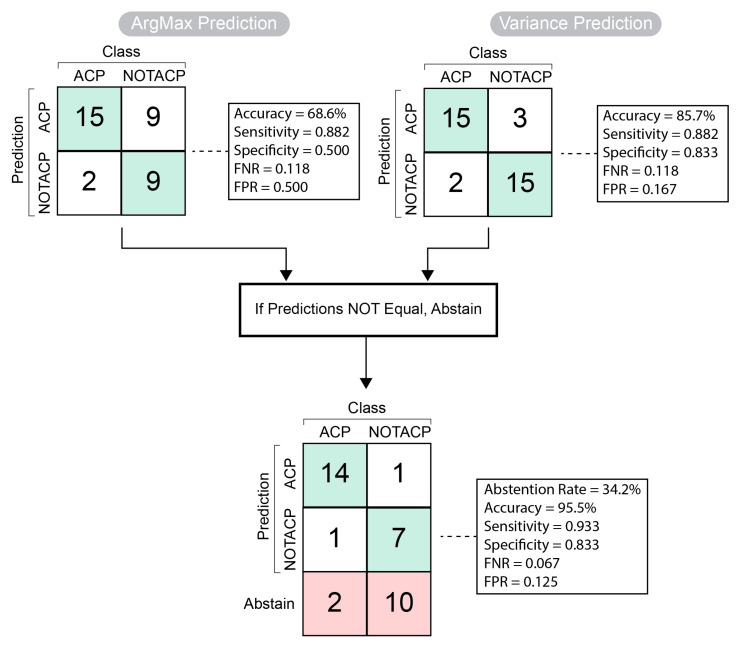
Abstention algorithm components and performance. Prediction by argument maximum is presented in the top-left and prediction by variance, each with associated performance metrics. The final prediction with abstentions is presented at the bottom.

## Data Availability

The data presented in this study are available upon request from the corresponding author. The data are not publicly available due to patient privacy reasons.

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
