# Peer review of "Uncertainty-Aware Deep Learning Classification of Adamantinomatous Craniopharyngioma from Preoperative MRI"

_diagnostics, 2023, doi:10.3390/diagnostics13061132_

Round 1
Reviewer 1 Report
This submission addresses the diagnosis of adamantinomatous craniopharyngioma (ACP).
The work is interesting but many concerns need to be carefully considered in the revised version.
1) Abstract: the details about the analyzed datasets should be summarized to help the reader.
2) All acronyms should be introduced throughout the text.
3) Section 1: These review papers would be helpful for improving the context of AI applications to brain tumor MRI analysis:
- Xie, Y., Zaccagna, F., Rundo, L., Testa, C., Agati, R., Lodi, R., ... & Tonon, C. (2022). Convolutional neural network techniques for brain tumor classification (from 2015 to 2022): Review, challenges, and future perspectives. Diagnostics, 12(8), 1850.
- Jaju, A., Yeom, K. W., & Ryan, M. E. (2022). MR imaging of pediatric brain tumors. Diagnostics, 12(4), 961. DOI: 10.3390/diagnostics12040961
4) Section 1: Too many methodological details are provided in Introduction. Perhaps, a separate Background section might be useful.
5) To some extent, the paper seems like a report (as also stated by the Authors in the Introduction). For instance, too many technical details are given (like names of the scripts ‘run_train_mode_connectivity.sh’, ‘run_swag_MRI.sh’).
6) Will the Authors make the source code available on GitHub? The link does not correctly work.
7) The novelty and the comparisons with respect to the previously published work by the Authors (Ref. [3]) need to be clarified. Please take care of this critical point.
For instance, the statement "The calibrated classifier underperformed our previously published results, indicating that the original model was overfit." has to be better explained.
8) The methodological details about the used machine learning models should be given and discussed.
9) The latest developments in the field of uncertainty quantification should be introduced:
- Abdar, M., Fahami, M. A., Rundo, L., Radeva, P., Frangi, A. F., Acharya, U. R., ... & Nahavandi, S. (2022). Hercules: Deep Hierarchical Attentive Multilevel Fusion Model With Uncertainty Quantification for Medical Image Classification. IEEE Transactions on Industrial Informatics, 19(1), 274-285. DOI: 10.1109/TII.2022.3168887
- Shamsi, A., Asgharnezhad, H., Jokandan, S. S., Khosravi, A., Kebria, P. M., Nahavandi, D., ... & Srinivasan, D. (2021). An uncertainty-aware transfer learning-based framework for COVID-19 diagnosis. IEEE Transactions on Neural Networks and Learning Systems, 32(4), 1408-1417. DOI: 10.1109/TNNLS.2021.3054306
10) Future work on the usability and utility of the implemented model should be better detailed.
11) The English language needs to be carefully reviewed.
Reviewer 2 Report
The introduction is well-written and good to read. The authors may consider mentioning the recent efforts of explainable AI.
The authors claim to have developed a simple algorithm for a classification abstention mechanism. The text, however, is very technical. It is comprehensive only for users of deep learning and readers that are familiar with applied programs, such as, e.g., tensorflow. The term “simple algorithm” may not be appropriate, at least for readers with a pure medical background.
The proposed combination of two approaches for classification as an abstention mechanism for individual patients is intuitive and worthwhile. The quality of such an abstention mechanism relies on the qualities of the two approaches. If the approaches gain complementary information a combination is fruitful. The study demonstrated that the incorporation of variance as a criterion is worthwhile at least for their patient group. I wonder whether the variance is a good criterion because of the diversity of patients in the NOTACP group. The results may be an artifact that is produced by the selection of the NOTACP group and may have no relevance for clinical work. It is not obvious, how the approach contributes to overcoming the lack of model transparency in DL and increasing the acceptance of DL in clinical practice. The patient cohort is by far too small to support the conclusion of the manuscript. The results are interesting. Further investigation of the role of variance may be a worthwhile approach. I would not recommend proposing the approach for clinical diagnosis even with visual design interfaces to improve usability and utility.
I found some typos, checking the text with an automatic tool, such as, e.g., Grammarly, may be worthwhile.
Figure 5: Please indicate which performance metric (in the top part) belongs to which approach.
Round 2
Reviewer 2 Report
I thank the authors for revising the manuscript.